# Socioeconomic Differences in Occupational Health Service Utilization and Sickness Absence Due to Mental Disorders: A Register-Based Retrospective Cohort Study

**DOI:** 10.3390/ijerph17062064

**Published:** 2020-03-20

**Authors:** Jaakko Harkko, Hilla Sumanen, Olli Pietiläinen, Kustaa Piha, Minna Mänty, Tea Lallukka, Ossi Rahkonen, Anne Kouvonen

**Affiliations:** 1Faculty of Social Sciences, University of Helsinki, 00014 Helsinki, Finland; hilla.sumanen@helsinki.fi (H.S.); anne.kouvonen@helsinki.fi (A.K.); 2Department of Public Health, Faculty of Medicine, University of Helsinki, 00014 Helsinki, Finland; olli.pietilainen@helsinki.fi (O.P.); kustaa.piha@helsinki.fi (K.P.); minna.manty@helsinki.fi (M.M.); tea.lallukka@helsinki.fi (T.L.); ossi.rahkonen@helsinki.fi (O.R.); 3South Eastern Finland University of Applied Sciences, 48220 Kotka, Finland; 4City of Vantaa, Department of strategy and research, 01030 Vantaa, Finland; 5Research Institute of Psychology, SWPS University of Social Sciences and Humanities, 53-238 Wroclaw, Poland; 6Administrative Data Research Centre–Northern Ireland, Centre for Public Health, Queen’s University Belfast, Belfast BT12 6BA, UK

**Keywords:** mental disorders, occupational health service, sickness absence, health inequalities

## Abstract

Occupational health service (OHS) is the main provider of primary care services for the working population in Finland. We investigated whether socioeconomic differences in the utilization of OHS predict sickness absence (SA) due to mental disorders. We used register linkage data covering the employees of the City of Helsinki aged 18–34 years (*N* = 6545) and 35–54 years (*N* = 15,296) from 2009 to 2014. The outcome was medically certified long-term (over 11 days) SA due to mental disorders. Cox regression analyses were performed to obtain hazard ratios (HR) and their 95% confidence intervals (CIs). Employees with low socioeconomic position (SEP) used OHS more frequently. The number of OHS visits independently predicted SA due to mental disorders. HRs were 1.59 (95% CI 1.35, 1.86) for those with frequent visits and 1.73 (95% CI 1.30, 2.29) for those with a clustered visit pattern among 18–34 year old employees; and 1.46 (95% CI 1.18, 1.81) and 1.41 (95% CI 1.14, 1.74) among 35–54 year old employees, respectively. In both age groups, lower education and routine non-manual worker position indicated the highest probability of SA. Low SEP predicts both high OHS utilization and subsequent SA due to mental disorders. Medical records may be used to accurately predict future SA, and the results indicate that preventive measures should be targeted particularly to younger employees with lower SEP.

## 1. Introduction

In western countries, mental disorders are among the most important causes impeding full labor market participation [1]. In Finland, for example, mental disorders are now the leading diagnostic cause for sickness absence (SA) [2]. Despite a decrease in overall SA prevalence over the last decades, SA due to mental disorders has increased [2]. Socioeconomic differences in both all-cause and mental disorder specific SA have remained large [3]. Although the evidence and official recommendations suggest an increased focus on prevention of mental disorders and related work disability, both the increasing trend in overall levels of and sustained socioeconomic differences in mental health related SA suggest that timely access to adequate treatment and lack of adequate provision of the prevention of mental health related hazards still constitute a major challenge [1].

In the Finnish occupational health service (OHS) system all employees are entitled to employer-provided OHS that is mandated by the legislation. Most employers purchase state-subsidized easy to access and free primary care services for their employees on top of the statutory preventive OHS that include services such as work-related health surveillance. In Finland, occupational health services provides primary care services to employees in a way that is comparable to primary care in international comparisons in substance but differs in that the service is targeted only at the working population. Thus, OHS is one potential site for promoting effective workplace interventions for preventing SA due to mental disorders. The utilization of health care services is a known contributor to inequalities in health [4,5], and previous study from Finland found that frequent attendance at OHS is associated with subsequent SA [6], which in turn may lead to disability pension [7,8,9]. A gap in knowledge remains about the role of OHS utilization in the processes leading to health inequalities. The socioeconomic distribution of frequent OHS attendance is not known [6] and this knowledge is needed to provide information for planning preventive actions through targeted interventions [10]. Frequent OHS visits may also be associated with socioeconomically differentiated levels of SA [11,12,13] and also existing sickness burden, as previous studies have shown recurrent nature of both all-cause and mental ill-health related SA [14,15]. Processes leading to SA due to mental disorders are complex and the development of preventive interventions and optimal targeting of services requires more detailed information on how socioeconomic position (SEP), OHS and SA are connected through investigating them in a single study setting with a specific interest given to differences between age groups. In OHS, one particular challenge is how to identify those at high risk at an early stage using commonly available administrative health care data.

This study used linked register data to examine how socioeconomic factors together with the primary care utilization are associated with subsequent SA due to mental disorders. More specifically, we tested the following hypotheses (1) high frequency of visits to OHS primary care predicts medically certified SA due to mental disorders, (2) socioeconomic differences in OHS utilization and previous SA partially explain these differences and (3) socio-economic differences in SA are in turn partially explained by higher OHS utilization among those with lower SEP and by more prevalent prior SA found in lower SEP groups [16].

## 2. Materials and Methods

### 2.1. Study Population

This is a retrospective register-based cohort study. The study is a part of the Helsinki Health Study, which is an ongoing cohort study that covers the all the City of Helsinki employees [17]. The City of Helsinki is Finland’s largest employer with c. 38,000 employees annually. The City operates on a variety of public services ranging from healthcare, education, social welfare services to public transport, construction and technical services. There are hundreds of different occupational titles, from manual work to administrative employees and professionals. All employees have an easy and equal access to OHS without any fee. The City of Helsinki applies the same OHS and SA policies to all of its employees, and these policies have remained relatively stable during the study period [16].

In this study, the data were retrieved from OHS register for those employees with at least three OHS visits during the years 2009–2014, which is c. 60% of all employees. Employees aged 18–34 years were considered younger employees (*N* = 6545) and employees aged 35–54 years as older employees (*N* = 15,196). The age grouping was set in accordance with the European Commission’s Eurostat who categorize employees aged 15–34 as young employees [17].

### 2.2. Outcome

The outcome was operationalized as a medically certified sickness allowance due to mental disorders extending over 11 calendar days. Sickness allowance is awarded by the Social Insurance Institution of Finland (SII) if the SA spell exceeds ten consecutive working days of work incapacity [18] The SII’s register thus covers all sickness allowance spells extending over 11 calendar days for all working age permanent residents of Finland. The outcome covered all SA spells with F-diagnoses in the ICD-10 coding scheme. Time-to-event was measured from the index OHS visit until the date of the event occurrence, or the end of 365 days of follow-up. We set the follow-up time to begin 15 days after the OHS visit as the preliminary analyses indicated a high time-dependence of the OHS variables below the 15 days threshold, and this association is likely to relate to the medical certification policy of the employer, namely that SA is often granted by the OHS general practitioner, rather than any substantial time pattern related to the objectives of this study.

### 2.3. Utilization of OHS

We used two indicators to reflect the patterns of utilization of OHS primary care services. The measures include all-cause service utilization and were not limited to mental health causes. The first measure is the total number of visits to OHS primary care services within one year to the index OHS visit, including the index visit. The number of visits was categorized into three groups: 1–3, 4–6 and 7 visits or more. The last category reflects so called frequent attenders, namely those in the highest decile of the number of consultations [9,10]. The second measure is the clustered visits preceding the index OHS visit and indicates a pattern of OHS visits where there are less than 14 calendar days between each visit. The number of visits extending 4 were categorized as 4. The values of the predictors were extracted in relation to each OHS visit uniquely. The data were derived from the OHS electronic medical records.

### 2.4. Socioeconomic Position

Two indicators of SEP were used to examine the association between social and economic factors, OHS utilization and SA. Firstly, educational attainment, obtained from Statistics Finland’s registry, was classified into four levels: higher education (a Master’s or a doctoral degree), upper secondary (a Bachelor’s degree), lower secondary (upper-secondary school, vocational school) and compulsory education (comprehensive school, equivalent of 9 years of schooling). Secondly, occupational class, obtained from the City of Helsinki employment records, was measured with four categories: “managers and professionals” (e.g., teachers with master’s degree and physicians), “semi-professionals” (e.g., nurses and foremen), “routine non-manuals” (e.g., clerical employees and child care workers) and “manual workers” (e.g., technical and cleaning staff). Information of both SEP variables was collected on an annual basis, thus reflecting the person’s SEP in the year of the index OHS visit.

### 2.5. Prior SA Spells

Prior SA spells due to mental disorders were included to control for the effect arising from recurrence of SA. As such, this measure overlaps with the outcome measure, but we included only SA spells starting before the index OHS visit. The variable was categorized into three groups: No prior SA, SA spells within 2 years before the index OHS visit, and SA spells within over 2 years before the index visit.

### 2.6. Statistical Analysis

Hazard ratios (HRs) with their 95% confidence intervals (95% CIs) for SA were calculated using the Cox regression procedure. We picked three observations (OHS visits) per employee, which were selected by random from the full OHS data by the statistical software. The data were set up in multiple occurrence format according to PWP-GT (gap time) model principles [19,20]. Time-to-event was measured for each of the three OHS visits individually, and the follow-up time began at zero for each observation. We obtained shared coefficients for all observations with unique baseline hazards according to the order of the observation. Intragroup correlation was allowed by calculating robust standard errors [19,20]. The baseline characteristics describing the study population in Table 1 relate to the employee’s first OHS visit.

Statistical analyses were stratified by age groups 18–34 and 35–54 to achieve age specific estimates. For the results presented in Table 2, the aim was to examine the predictive quality of OHS utilization patterns. To do so, we sequentially introduced the variables in subsequent regression models. First, the OHS variables were entered individually (Model 1). Educational attainment and occupational class (Model 2) and prior SA (Model 3) were introduced first separately and then together in the final model (Model 4). For the results presented in Table 3, the models were fitted to examine how OHS utilization patterns relate to socioeconomic differences in SA. Similarly to the previous analysis, educational attainment and occupational class were first introduced into the models individually (Model 1), and then controlled for OHS utilization variables (Model 2) and prior SA (Models 3) first separately, and finally all variables were included in the same model (Model 4). We did not find statistically significant interactions between gender and the index OHS variables and all analyses were thus adjusted for gender. The baseline characteristics by gender are provided in Appendix A.

The proportional hazards assumption was tested with the test of Schoenfeld residuals, which showed proportionality for all variables except “Total OHS visits” in the sub-group of older employees. Further investigation with including the time-variable interaction into the models led us to conclude that this non-proportionality reflects a substantive time pattern and the estimates are presented for both as direct and time-varying effects. The former represents the HR of the variable at the beginning of the follow-up, and the latter represents an increase of hazard every month thereafter.

Sensitivity analyses were run first by comparing the results with an analysis of the current data with 6 months follow-up time, and by reproducing the estimates with a dataset with three different random OHS visits per person.

### 2.7. Ethical Considerations

The study follows the Helsinki Health Study (HHS) protocol in line with the University of Helsinki’s guidelines and the EU and Finnish data legislation. The ethics committees of the Department of Public Health, the University of Helsinki and the health authorities of the City of Helsinki have approved the HHS study. The City of Helsinki and register holders have given permission for data linkage.

## 3. Results

### 3.1. Baseline Characteristics

Among the 6545 younger employees and 15,196 older employees in the study population, 6.0% of young employees and 5.4% of older employees had been granted SA due to mental disorders in one year after the OHS visit (Table 1). The number of OHS visits was ca. 2550 per thousand younger employees and 3000 per thousand older employees during the year preceding their first OHS visit. The total number of OHS visits did not differ between the genders, but women had more SA due to mental disorders both before and after the index OHS visit. The gender difference in SA was more pronounced among the younger employees. Employees with less education and lower occupational class had both higher rates of OHS attendance and SA. In both age groups SA spells were the most prevalent among those with lower secondary education, and in routine non-manual workers.

SA spells were more prevalent among those with higher frequency of OHS use. For example, those with seven or more visits (so called frequent attenders) had 2–3 times more often SA spells due to mental disorders compared to those who had the lowest number of visits. A similar pattern was found for the second primary care visit variable. The data indicate a high propensity for the recurrence of SA due to mental disorders. Of those who had had SA due to mental disorders in the two years before the index OHS visit, approximately one in five experienced such an event also during the follow-up, while the corresponding figure for those with no prior SA were only 3–4 percent.

### 3.2. Main Results

Both the high frequency of OHS visits and clustered OHS visits predicted SA (Table 2). Gender adjusted HRs for the frequent visitors group compared with those with the lowest attendance was 2.34 (95% CI 2.01, 2.73) for the younger employees and 2.48 (2.01–3.06) with a 3% increase in risk each month for the older employees, respectively. The number of clustered OHS visits predicted SA in a similar dose-response manner. The attendance patterns predicted later SA largely independently of SEP measures. Controlling for education and occupational class reduced the difference only slightly. However, as expected, prior SA due to mental disorders had a considerable effect on the associations. Adjusting for all variables lowered the HRs for those with seven or more OHS visits to 1.59 (95% CI 1.35–1.62) for the younger employees and 1.46 (95% CI 1.18, 1.81) with a 4% increase in risk each month for the older employees, respectively. Notably, clustered OHS visits indicated better predictive qualities for the younger employees, whereas the more traditional frequency of attendance had better predictive qualities for the older employees, after accounting for the time–variable interaction.

The results reflecting SA by education and occupational class (Table 3) showed that both SEP measures predicted SA in the first, gender adjusted model. The risk for SA was highest for those with basic and lower secondary education. Compared to the higher education group, HR for the younger employees with only compulsory education was 2.02 (95% CI 1.58, 2.59), and the corresponding HR for the older employees was 1.49 (95% CI 1.28, 1.75). In the comparison between occupational classes, compared to managers and professionals, routine non-manual workers had the highest risk, and this risk (HR 1.77, 95% CI 1.49, 2.10) was higher among the younger employees than the older employees (HR 1.53, 95% CI 1.37, 1.70). Accounting for prior OHS utilization and prior SA reduced the HR estimates, the latter more pronouncedly, indicating that both OHS utilization histories and previous SA partially, but not fully, explain the occupational differences in SA. In the full model, for those with only compulsory education, HRs was 1.60 (95% CI 1.24, 2.06) for the younger employees, and 1.13 (95% CI 0.97, 1.33) for the older employees, respectively. Corresponding HRs for routine non-manual workers were 1.41 (95% CI 1.19, 1.68) and 1.14 (95% CI 1.02, 1.28), respectively.

The results of the sensitivity analyses indicated robustness of our findings.

## 4. Discussion

In the present register-based study in a large cohort of public sector employees, we found that both occupational health service utilization and sickness absence due to mental disorders followed consistent socioeconomic patterns. High frequency of visits to OHS primary care and medically certified SA due to mental disorders were both concentrated in the lower SEP groups, especially in those with lower educational qualifications and in routine non-manual jobs. The main findings were that the younger employees had slightly fewer OHS visits, but more SA due to mental disorders as compared to the older employees, and that socioeconomic differences in SA were more marked among the younger employees. Frequent OHS attendance was more prevalent for those at risk of SA due to mental disorders, and frequent attendance predicted SA independently of SEP and prior SA. We further found that a considerable proportion of the effect of low SEP on SA due to mental disorders is mediated through OHS utilization and prior SA, indicating higher burden of disease among lower occupational classes. Yet, the direct effect of occupational class remained even after adjusting for these factors, particularly among younger employees.

There were differences between the age groups in how well different measures, i.e., OHS patterns, predicted SA. In line with previous studies [6,7], we showed that the employees with a risk of subsequent SA might be identified by their frequent OHS utilization. The present study further showed that frequent attendance may be captured by different utilization pattern categories: the traditional frequent attendance measure as the cumulative visits within a year, but also by the clustered patterns of OHS visits. The latter may be more sensitive to early intervention whereas the more traditional frequent attendance measure may be used only after a rather long measurement time. The novelty of our approach lies in combining findings on socioeconomic and health service related determinants on mental health related work disability in a single explanatory framework. Both of these categories of determinants have consistently been shown to predict SA [3,4,5,6]. In this study we were able to study these factors together with high quality register data. As a limitation, our data did not include factors related to preventive workplace interventions such as enhancing employee control, promoting physical activity or therapeutic interventions [21]. We did not test whether the effects of the primary independent variables on the outcome were mediated by the covariates and this may be considered as a further limitation of this study.

This study was based on a large number of employees of the City of Helsinki. The cohort included all 18–54 year old employees with three or more OHS visits, and over 21,000 employees with individual OHS utilization patterns were included in the data. The OHS data were linked to data from employer’s and SII’s registers using national ID numbers, practically without loss to follow-up. The registers constitute a reliable and comprehensive data source on the frequency of visits and SA with no inaccuracy related to self-reporting. Using SA as an outcome may be considered as a strength of this study as SA indicates a medically relevant work disability and it is also considered to be a reliable health indicator [22,23]. From the policy perspective, a further benefit of using SA as an outcome measure is that SA has been found to be a predictor for later disability pension and mortality [8,9]. Our results may thus be used in planning preventive programs beyond the current study context. By linking data on education and occupational class, we could address the issue of socioeconomic differences in SA in the OHS context, which was highlighted as a gap in knowledge in an earlier study [6].

However, there are limitations to the study. As a measure of ill health, SA present a problem of a degree of under-coverage as the vast majority of those with a health problem do not claim sickness allowance [23,24]. We had no other information about the prevalence and severity of health problems of the employees, which may have caused unobserved health selection to the adverse predictor groups. We could not differentiate between socioeconomic factors related to differences in health, such as health behaviors, benefit seeking or adverse working conditions. Although universal and free access to OHS may lessen the evidenced inequality in access to health care in other settings [4,5], we cannot ascertain the adequacy of treatment in OHS in relation to the health concerns among different SEP groups. A further limitation is that we used only three randomly selected OHS visits per employee and the information regarding longitudinal dynamics of OHS visits was lost. For example, we did not explicitly model the time intervals of visit sequences. Further studies should address these temporal issues.

Caution is also needed when generalizing the results regarding SEP. Only OHS clients were included in the data and the results therefore reflect the effects of SEP among this clientele and not among all employees. The latter includes those with no health concerns and is likely to include more highly educated and those in higher occupational positions [25]. Furthermore, despite Helsinki being a large public sector workplace with wide-ranging operations, generalizing the results to other organizations or social security settings must be done cautiously.

Despite these limitations, our results indicate important issues for policy makers as well as for occupational health and primary care specialists. First, we argued that our approach of using date-specific OHS visit data to detect different types of OHS patterns that indicate subsequent SA may provide itself useful as the provided robust and specific estimates outperform the studies only using annual or such aggregate data. Second, our results may invite researchers from other settings to conduct research and to develop real-life applications for those in the highest need by simultaneously monitoring both socioeconomic and health service related predictors of SA.

## 5. Conclusions

Our findings strongly suggest that both the frequent OHS utilization and SA due to mental disorders are concentrated in lower SEP groups, and that SA due to mental disorders were more prevalent in these groups despite the universal coverage of primary care services in the occupational health setting. In this study we explored the reasons for socioeconomic differences in the patterns of OHS utilization and SA due to mental disorders. We found that there was a group of young lower SEP employees with high risk of SA due to mental disorders. Further studies are warranted to explore how the needs of these workers are met by the current arrangements of work and working conditions and by the services aiming at prevention of health hazards related to mental disorders. The use of routinely collected data may prove to be a powerful resource to help OHS professionals to identify high risk employees for implementing the effective individual level or workplace interventions tackling mental disorders.

## Figures and Tables

**Table 1 ijerph-17-02064-t001:** Distributions of study variables, means of occupational health service (OHS) visits at baseline and sickness absence (SA) at baseline and during the follow-up among the employees of the City of Helsinki, by age group.

	18–34 Years				35–54 Years			
	Subjects	OHS Visits, Baseline	SA, Baseline	SA, Follow-Up	Subjects	OHS Visits, Baseline	SA, Baseline	SA, Follow-Up
	*N* (%)	1/1000	*N* (%)	*N* (%)	*N* (%)	1/1000	*N* (%)	*N* (%)
All	6545 (100.0)	2554	404 (6.2)	391 (6.0)	15196 (100.0)	3024	1223 (8.0)	828 (5.4)
Time-to-event in months, total (average)	76343 (11.7)				177368 (11.7)			
Study variables								
Gender								
Men	1517 (23.2)	2562	63 (4.2)	49 (3.2)	3632 (23.9)	2983	204 (5.6)	128 (3.5)
Women	5028 (76.8)	2552	341 (6.8)	342 (6.8)	11564 (76.1)	3037	1019 (8.8)	700 (6.1)
Education								
Higher education	982 (15.0)	2126	35 (3.6)	38 (3.9)	3011 (19.8)	2424	173 (5.7)	129 (4.3)
Upper secondary	1800 (27.5)	2339	94 (5.2)	98 (5.4)	4623 (30.4)	2849	377 (8.2)	260 (5.6)
Lower secondary	3062 (46.8)	2741	225 (7.3)	209 (6.8)	5899 (38.8)	3333	540 (9.2)	346 (5.9)
Compulsory education	701 (10.7)	2886	50 (7.1)	46 (6.6)	1663 (10.9)	3497	133 (8.0)	93 (5.6)
Occupational class								
Managers or professionals	1166 (17.8)	2177	48 (4.1)	50 (4.3)	3812 (25.1)	2502	228 (6.0)	156 (4.1)
Semi-professionals	1531 (23.4)	2299	82 (5.4)	77 (5.0)	3729 (24.5)	2857	297 (8.0)	208 (5.6)
Routine non-manual workers	2754 (42.1)	2728	223 (8.1)	214 (7.8)	5028 (33.1)	3313	518 (10.3)	339 (6.7)
Manual workers	1094 (16.7)	2875	51 (4.7)	50 (4.6)	2627 (17.3)	3464	180 (6.9)	125 (4.8)
Total OHS visits in 1 year								
1–3	5163 (78.9)	1611	228 (4.4)	260 (5.0)	10813 (71.2)	1732	591 (5.5)	475 (4.4)
4–6	975 (14.9)	4708	91 (9.3)	77 (7.9)	2971 (19.6)	4720	335 (11.3)	212 (7.1)
7 or more	407 (6.2)	9354	85 (20.9)	54 (13.3)	1412 (9.3)	9347	297 (21.0)	141 (10.0)
Clustered OHS visits^1^								
1	4933 (75.4)	2127	281 (5.7)	253 (5.1)	11631 (76.5)	2624	850 (7.3)	587 (5.0)
2	1134 (17.3)	3253	59 (5.2)	83 (7.3)	2553 (16.8)	3781	215 (8.4)	158 (6.2)
3	325 (5.0)	4662	36 (11.1)	33 (10.2)	688 (4.5)	5058	97 (14.1)	57 (8.3)
4 or more	153 (2.3)	6667	28 (18.3)	22 (14.4)	324 (2.1)	7077	61 (18.8)	26 (8.0)
F-diag prior to OHS visit								
No	5738 (87.7)	2402	0 (0.0)	249 (4.3)	12175 (80.1)	2807	0 (0.0)	372 (3.1)
> 2 years to OHS visit	403 (6.2)	3050	0 (0.0)	63 (15.6)	1798 (11.8)	3360	0 (0.0)	186 (10.3)
0–2 years to OHS visit	404 (6.2)	4213	404 (100.0)	79 (19.6)	1223 (8.0)	4689	1223 (100.0)	270 (22.1)

^1^ Clustered OHS visits indicate the number of OHS visits with less than 14 calendar days between each visit.

**Table 2 ijerph-17-02064-t002:** Sickness absence due to mental disorders among the employees of the City of Helsinki by age group and OHS utilization, hazard ratios (HRs; 95% CIs).

	Model 1 ^1^	Model 2 ^2^	Model 3 ^3^	Model 4 ^4^
Age: 18–34 years				
Total OHS visits in 1 year				
1–3	1.00	1.00	1.00	1.00
4–6	1.52 (1.33–1.74)	1.47 (1.29-1.69)	1.34 (1.17–1.53)	1.32 (1.15–1.51)
7 or more	2.34 (2.01–2.73)	2.17 (1.85–2.53)	1.66 (1.42–1.95)	1.59 (1.35-1.86)
Clustered OHS visits				
1	1.00	1.00	1.00	1.00
2	1.38 (1.20–1.58)	1.34 (1.16–1.54)	1.39 (1.21–1.60)	1.36 (1.18–1.56)
3	1.89 (1.52–2.35)	1.81 (1.45–2.26)	1.71 (1.38–2.14)	1.69 (1.35–2.10)
4 or more	2.41 (1.84–3.15)	2.21 (1.68–2.91)	1.83 (1.39–2.42)	1.73 (1.30–2.29)
Age: 34–54 years				
Total OHS visits in 1 year				
1–3	1.00	1.00	1.00	1.00
4–6	1.35 (1.10–1.67)	1.33 (1.09-1.64)	1.06 (0.87–1.31)	1.06 (0.86–1.30)
7 or more	2.48 (2.01–3.06)	2.41 (1.95–2.97)	1.48 (1.19-1.83)	1.46 (1.18–1.81)
4–6 # time ^5^	1.03 (1.01–1.06)	1.03 (1.01–1.06)	1.04 (1.01–1.07)	1.04 (1.01–1.07)
7 or more # time ^5^	1.03 (1.00–1.06)	1.03 (1.00–1.06)	1.04 (1.01-1.07)	1.04 (1.01–1.07)
Clustered OHS visits				
1	1.00	1.00	1.00	1.00
2	1.37 (1.24–1.51)	1.34 (1.21–1.48)	1.29 (1.17–1.43)	1.28 (1.16–1.42)
3	1.88 (1.60–2.20)	1.83 (1.56–2.15)	1.44 (1.23–1.69)	1.42 (1.21–1.67)
4 or more	2.04 (1.66–2.50)	1.97 (1.60–2.42)	1.43 (1.16–1.77)	1.41 (1.14–1.74)

^1^ Model 1 = Predictor + gender. ^2^ Model 2 = Predictor + gender + education + occupational class. ^3^ Model 3 = Predictor + gender + prior SAs. ^4^ Model 4 = Predictor + gender + education + occupational class + prior SAs. ^5^ Interaction term for time and variable.

**Table 3 ijerph-17-02064-t003:** Sickness absence due to mental disorders among the employees of the City of Helsinki by age group and socioeconomic position, HRs (95% CIs).

	Model 1 ^1^	Model 2 ^2^	Model 3 ^3^	Model 4 ^4^
Age: 18–34				
Education				
Higher education	1.00	1.00	1.00	1.00
Upper secondary	1.35 (1.09–1.67)	1.30 (1.05–1.61)	1.25 (1.01–1.55)	1.23 (0.99–1.52)
Lower secondary	2.02 (1.66–2.46)	1.82 (1.50–2.22)	1.69 (1.39–2.06)	1.59 (1.30–1.94)
Compulsory education	2.02 (1.58–2.59)	1.75 (1.36–2.25)	1.74 (1.35–2.23)	1.60 (1.24–2.06)
Occupational class				
Managers or professionals	1.00	1.00	1.00	1.00
Semi-professionals	1.07 (0.88–1.31)	1.03 (0.85–1.26)	1.00 (0.82–1.22)	0.98 (0.80–1.19)
Routine non-manual workers	1.77 (1.49–2.10)	1.61 (1.35–1.91)	1.49 (1.25–1.77)	1.41 (1.19–1.68)
Manual workers	1.14 (0.89–1.44)	1.00 (0.78–1.27)	1.07 (0.84–1.36)	0.98 (0.77–1.25)
Age: 35–54				
Education				
Higher education	1.00	1.00	1.00	1.00
Upper secondary	1.35 (1.19–1.53)	1.25 (1.11–1.42)	1.18 (1.04–1.33)	1.13 (1.00–1.28)
Lower secondary	1.48 (1.32–1.67)	1.27 (1.12–1.43)	1.24 (1.10–1.39)	1.14 (1.01–1.29)
Compulsory education	1.49 (1.28–1.75)	1.21 (1.03–1.42)	1.27 (1.08–1.48)	1.13 (0.97–1.33)
Occupational class				
Managers or professionals	1.00	1.00	1.00	1.00
Semi-professionals	1.23 (1.09–1.39)	1.15 (1.02–1.30)	1.09 (0.97–1.23)	1.05 (0.93–1.19)
Routine non-manual workers	1.53 (1.37–1.70)	1.31 (1.17–1.46)	1.23 (1.10–1.38)	1.14 (1.02–1.28)
Manual workers	1.26 (1.10–1.45)	1.04 (0.90–1.20)	1.15 (1.00–1.32)	1.03 (0.90–1.19)

^1^ Model 1 = Predictor + gender. ^2^ Model 2 = Predictor + gender + Total OHS visits in 1 year + Clustered OHS visits. ^3^ Model 3 = Predictor + gender + prior SAs. ^4^ Model 4 = Predictor + gender + Total OHS visits in 1 year + Clustered OHS visits + prior SAs.

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
