# Peer review of "Socioeconomic Differences in Occupational Health Service Utilization and Sickness Absence Due to Mental Disorders: A Register-Based Retrospective Cohort Study"

_ijerph, 2020, doi:10.3390/ijerph17062064_

Round 1

Reviewer 1 Report

The authors conducted a retrospective study investigating whether socioeconomic differences in the utilization of health care services predict sickness absence due to mental disorders.  Overall, this is an important study and a well-written manuscript.  Some suggestions/questions follow that could be considered.

Introduction

  1. Line 39 and 40 – sentence needs rewording for clarity
  2. Line 42 and 43 – why has timely access to treatment been a challenge?
  3. Line 48 – any mental health professionals other than psychologists?
  4. Might provide more background regarding the current state of the literature on this general topic and what gaps this research fills

Materials and methods

  1. Lines 77 - 79 – why did the study population not include subjects over age 54?
  2. Lines 81 – 83 – please clarify that sickness absence was defined as being of work for > 11 days?
  3. Lines 103 – 113 – it would be helpful to add education equivalents for US and perhaps readers from other countries who use different terminologies than Finland.  For example, what is the US equivalent of “comprehensive school?”  Also – how was the occupational classification categorization determined?  For example, why is a teacher considered a professional and a nurse considered semi-professional?   From a US reader standpoint – these classifications seem dated.

Discussion

  1. Do the authors want to say more about potential reasons for high frequency primary care visits and higher utilization of SA concentrating among lower SEP groups and/or reference some existing literature about this?

Conclusions

  1. Line 287 & 288 – how did the investigators reach the conclusion that “We found that there are group of young lower SEP 287 employees with high SA due to mental disorders that traditional services do not reach.”?  Didn’t this group actually receive a lot of services?  Or is it that the services were ineffective?

Author Response

Response to Reviewer 1 Comments

Comment 1: The authors conducted a retrospective study investigating whether socioeconomic differences in the utilization of health care services predict sickness absence due to mental disorders.  Overall, this is an important study and a well-written manuscript.  Some suggestions/questions follow that could be considered.

Response 1: We thank the reviewer for this positive feedback.

Introduction

Comment 2: Line 39 and 40 – sentence needs rewording for clarity

Response 2: Thank you for noting this. We have rephrased the sentence as follows: ”Socioeconomic differences in both all-cause and mental disorder specific SA have remained large [3].” (Page 1)

Comment 3:  Line 42 and 43 – why has timely access to treatment been a challenge?

Response 3: Thank you for pointing this out. We have rephrased the sentence as follows: ”Although the evidence and official recommendations suggest an increased focus on prevention of mental disorders and related work disability, both the increasing trend in overall levels of prevalence and sustained socioeconomic differences in mental health related SA suggest that timely access to adequate treatment and lack of adequate provision of the prevention of mental health related hazards still constitute a major challenge.” (page 1)

Comment 4:  Line 48 – any mental health professionals other than psychologists?

Response 4: Thank you for noting this. We deleted the sentence to provide clarity and to avoid confusion.

Comment 5: Might provide more background regarding the current state of the literature on this general topic and what gaps this research fills

Response 5: Thank you for pointing this out. We have given effort to better describe our motivation behind the study by providing additional references and by better description of our research aims. We have revised the following paragraph in the introduction (pages 1-2).

Thus, OHS is one potential site for promoting effective workplace interventions for preventing SA due to mental disorders. The utilisation of health care services is a known contributor to inequalities in health [7, 8], and previous study from Finland found that frequent attendance at OHS is associated with subsequent SA [9], which in turn may lead to disability pension [10-12]. A gap in knowledge remains about the role of OHS utilisation in the processes leading to health inequalities. The socioeconomic distribution of frequent OHS attendance is not known [9], and this knowledge is needed to provide information for planning preventive actions through targeted interventions [13]. Frequent OHS visits may also be associated with socioeconomically differentiated levels of existing sickness burden, as previous studies have shown recurrent nature of both all-cause and mental ill-health related SA [14, 15]. Processes leading to SA due to mental disorders are complex and the development of preventive interventions and optimal targeting of services requires more detailed information on how SEP, OHS and SA are connected through investigating them in a single study setting with a specific interest given to differences between age groups.

We also added the following references:

13 Robinson M, Fisher TF, Broussard K. Role of occupational therapy in case management and care coordination for clients with complex conditions. American Journal of Occupational Therapy. 2016 Mar 1;70(2):7002090010p1-6.

14 Laaksonen M, He L, Pitkäniemi J. The durations of past sickness absences predict future absence episodes. J Occup Environ Med. 2013 Jan;55(1):87-92.

15 Sumanen H, Pietiläinen O, Lahelma E, Rahkonen O. Short sickness absence and subsequent sickness absence due to mental disorders - a follow-up study among municipal employees. BMC Public Health. 2017 Jan 5;17(1):15. 

Materials and methods

Comment 6: Lines 77 - 79 – why did the study population not include subjects over age 54?

Response 6: The perspective of the prevention of SA among younger employees in one of the key motivations behind this manuscript. We chose to use the threshold to exclude the very oldest employees to avoid exposing our analyses to confounding related specifically to the late-working-life issues. An example of such issues is that disability pensions become a relevant risk category for work disability at the later stage of the working career and the survivor effect may cause bias to the estimates, i.e. the older employees may systemically differ from their younger peers because of the increasing incidence of disability pension.

Comment 7: Response 6: Lines 81 – 83 – please clarify that sickness absence was defined as being of work for > 11 days?

Response 7: Thank you for noting this. We have rephrased the definition as follows: ”The outcome was operationalised as a medically certified sickness allowance due to mental disorders extending over 11 calendar days. Sickness allowance is awarded by the Social Insurance Institution of Finland  (SII) if the SA spell exceeds ten consecutive working days of work incapacity [18]. ” (page 2)

We added the following reference to the description of sickness allowances:

18. Sickness allowance. Social insurance institution of Finland. https://www.kela.fi/web/en/sickness-allowance (accessed 14.3.2020)

Comment 8: Lines 103 – 113 – it would be helpful to add education equivalents for US and perhaps readers from other countries who use different terminologies than Finland.  For example, what is the US equivalent of “comprehensive school?”  Also – how was the occupational classification categorization determined?  For example, why is a teacher considered a professional and a nurse considered semi-professional?   From a US reader standpoint – these classifications seem dated.

Response 8: Thank you for indicating these important places for clarification. With regards to education, we have standardized the used terminology and replaced ”basic education” with ”compulsory education” throughout the text. We also added a clarification in the method section as follows: ”compulsory education (comprehensive school, equivalent of 9 years of schooling). ” (page 3). In Finland, mandatory education is free for all and the vast majority of children and young people go to public schools. As to occupational classification, we have used a standardised classification structure that follows a classification by Statistic Finland, is used in the City of Helsinki personnel register, and has been used consistently in scientific articles published by Helsinki Health Study since early 2000’s. For more detailed description, please refer to Aittomäki A. Social-class inequalities in ill health – the contribution of physical workload. Publications of Public Health M 195:2008. Helsinki: University of Helsinki, 2008.  (https://helda.helsinki.fi/handle/10138/20322 (see pages 100-1)).

On the specific question related to teachers and nurses, the answer is that the classification reflects the Finnish education system where teachers are required to have a five-year master’s degree whereas nurses are qualified through a three-year bachelor’s degree from the universities of applied sciences. We added a clarification ”(e.g. teachers with a master’s degree and physicians)” to the text. (page 3).

Discussion

Comment 9:  Do the authors want to say more about potential reasons for high frequency primary care visits and higher utilization of SA concentrating among lower SEP groups and/or reference some existing literature about this?

Response 9: Thank you for this important comment. We expect this to be due to a combination of factors. First we wish to highlight that one of the motivations behind this paper was to combine these factors into a single explanatory framework in the first place. Both SEP and health care utilisation have been studied as predictors of SA individually and the novelty of this study is to combine the perspectives. As to the actual question, we expect there to be common factors behind both predictors.  We have deliberately chosen not to cover this subject extensively as the scope of this register based study did not allow to directly observe such potentially explaining factors as employees’ job control and physical activity together with unevenly socially distributed mental health symptoms.

Conclusions

Comment 10:  Line 287 & 288 – how did the investigators reach the conclusion that “We found that there are group of young lower SEP employees with high SA due to mental disorders that traditional services do not reach.”?  Didn’t this group actually receive a lot of services?  Or is it that the services were ineffective?

Response 10: Thank you for bringing this important issue to our attention. We have rephrased this sentence as follows: “We found that there is a group of young lower SEP employees with a high risk of SA due to mental disorders. Further studies are warranted to explore how the needs of these workers are met by the current arrangements of work and working conditions and by the services aiming at prevention of health hazards related to mental disorders.” (page 11).

Reviewer 2 Report

Overall, I found that this paper was methodologically unobjectionable.  I think there is probably too much detail regarding the multiple models used for each outcome--it might be clearer to report the most detailed model, but to include interaction terms where there is a question about whether the effect of the primary independent variable on the outcome is mediated by one of the covariates or not. Alternatively, a regression model relating the independent variable to the covariates might be reported.  But these small concerns aside, the approach seems sound.  A larger concern pertains to the scientific significance of the findings.  First, it is fairly clear that the paper's results are not readily generalized to other populations outside of municipal workers in Helskinki. Second, to the extent that we can consider these findings in the context of other studies describing predictors of occupational disability, the study may not seem to add much that is not already known.  It suggests that more frequent primary care/occupational health visits are associated with a higher risk of absence from work. But that is widely known.  It also shows that lower socioeconomic status--here, measured crudely as educational attainment and occupational level--are associated with higher risk of work absence for mental health problems.   Again, however, this is widely known.  Thus, I think that the authors should place more emphasis in the discussion on articulating how this research adds to the body of general knowledge about these issues.  The study also examines the moderating effect of prior sick absences on the predictive power of OHS visits for further sick absences.  Unsurprisingly, it shows that past sick absences predict future sick absences.   Finally, I have doubts about the utility of the models reported in predicting sick absences in a way that would be helpful in targeting preventive interventions.  If that is truly the ambition of the paper, it would be useful to report a different measure that reflects the actual frequency of sick absences, such as a positive predictive value. 

Author Response

Response to Reviewer 2 Comments

Comment 1: Overall, I found that this paper was methodologically unobjectionable. 

Response 1: Thank you for this encouraging feedback.

Comment 2: I think there is probably too much detail regarding the multiple models used for each outcome--it might be clearer to report the most detailed model, but to include interaction terms where there is a question about whether the effect of the primary independent variable on the outcome is mediated by one of the covariates or not. Alternatively, a regression model relating the independent variable to the covariates might be reported.  But these small concerns aside, the approach seems sound. 

Response 2: Thank you for this comment. As the reviewer finds our approach sound, we decided to leave the structure of the analysis as it was. We added a notion to Discussion that we find not testing mediation as a limitation to discussion as follows: ”We did not test whether the effects of the primary independent variables on the outcome were mediated by the covariates and this may be considered as a further limitation of this study.” (page 10)

Comment 3: A larger concern pertains to the scientific significance of the findings.  First, it is fairly clear that the paper's results are not readily generalized to other populations outside of municipal workers in Helskinki.

Response 3: Thank you for this important note. While we have expressed this caution in the discussion (lines 276-277) we interpret the comment as a recommendation for more extensive clarifications across the manuscript. To do so,

First, we added a phrase clarifying the studied service ”In Finland, occupational health service provides primary care services for employees in a way that is comparable to primary care services in international comparisons in substance but differs in that it is targeted only at the working population.” (Page 1)

Second, we attempt to give a more precise definition of our research task in the introduction by the following two rephrasing: 1) ”The development of preventive interventions and optimal targeting of services requires more detailed information in this regard” was rephrased to ” Processes leading to SA due to mental disorders are complex and the development of preventive interventions and optimal targeting of services requires more detailed information on how SEP, OHS and SA are connected through investigating them in a single study setting with a specific interest given to differences between age groups.” (page2). 2) ”This study uses linked register data to examine socio-economic differences in the OHS utilisation and subsequent SA due to mental disorders.” was rephrased to ”This study uses linked register data to examine how socio-economic factors together with the primary care utilisation are associated with subsequent SA due to mental disorders”. (Page 2)

Third, we rephrased the last paragraph of discussion to sum up the refinements above . ”Despite these limitations, our results indicate important issues for policy makers as well as for occupational health and primary  care specialists. First, we argue that our approach of using date-specific OHS visit data to detect different types of OHS patterns that indicate subsequent SA may provide itself useful as the provided robust and specific estimates outperform the studies only using annual or such aggregate data. Second, our results may invite researchers from other settings to conduct research and to develop real-life applications for those in the highest need by simultaneously monitoring both socioeconomic and health service related predictors of SA.” (Page 11)

Comment 4: Second, to the extent that we can consider these findings in the context of other studies describing predictors of occupational disability, the study may not seem to add much that is not already known. It suggests that more frequent primary care/occupational health visits are associated with a higher risk of absence from work. But that is widely known. It also shows that lower socioeconomic status--here, measured crudely as educational attainment and occupational level--are associated with higher risk of work absence for mental health problems.   Again, however, this is widely known.  Thus, I think that the authors should place more emphasis in the discussion on articulating how this research adds to the body of general knowledge about these issues. 

Response 4: Thank you for pointing this out. We hope that we have answered these concerns in the previous response that gives a more detailed account of our motivation behind the study, i.e. that we aim to explore the socioeconomic and health service related determinants on work disability in a single explanatory framework. We have made an effort to emphasis to clarify the novelty of  this study throughout the text. We have also added the following sentence to Discussion: ”The novelty of our approach lies in combining findings on socioeconomic and health service related determinants on mental health related work disability in a single explanatory framework. Both of these categories of determinants have consistently been shown to predict SA [3, 7-9]. In this study we were able to study these factors together with high quality register data.” (Page 10)

Comment 5: The study also examines the moderating effect of prior sick absences on the predictive power of OHS visits for further sick absences.  Unsurprisingly, it shows that past sick absences predict future sick absences.   

Response 5: The comment relates to our Responses 3 and 4 above, and we hope that we have answered these questions in the answer 3. However, we wish to point out that the referred finding is as the matter of fact non-trivial in the context of this study. In terms of preventive interventions, a distinction between accumulation of sickness burden over longer time period and acute diseases is crucial and the results indicate that unadjusted frequent OHS utilisation variable may include effects from the both. We find this to be an interesting topic for research in the future.

Comment 6: Finally, I have doubts about the utility of the models reported in predicting sick absences in a way that would be helpful in targeting preventive interventions.  If that is truly the ambition of the paper, it would be useful to report a different measure that reflects the actual frequency of sick absences, such as a positive predictive value.

Response 7:We thank the reviewer for highligting this. First, we want to point out that the actual frequencies of sickness absence by predictive categories are presented in Table 1. The comment is likely targeted at the way we present our statistical estimates from the regression analyses. It is true that hazard ratios – a representation of risks on a multiplicative scale – may not allow immediate comparison of models in terms of case frequencies. However, the presentation of the actual frequencies in the descriptive part of the results and statistical estimates as hazard ratios is an established practice in this field of study, and we believe that the presented results provide a sufficient and reliable information base for the purpose of planning preventive interventions.